# Variants of *MIRNA146A* rs2910164 and *MIRNA499* rs3746444 are associated with the development of cutaneous leishmaniasis caused by *Leishmania guyanensis* and with plasma chemokine IL-8

**Tirza Gabrielle Ramos de Mesquita**[1,2], **José do Espírito Santo Junior**[3,4], **Thais Carneiro de Lacerda**[1,4], **Krys Layane Guimarães Duarte Queiroz**[2], **Cláudio Marcello da Silveira Júnior**[2], **José Pereira de Moura Neto**[5], **Lissianne Augusta Matos Gomes**[4], **Mara Lúcia Gomes de Souza**[2], **Marcus Vinitius de Farias Guerra**[1,2,6], **Rajendranath Ramasawmy**[1,2,4,6]*

1 Programa de Pós-Graduação em Medicina Tropical, Universidade do Estado do Amazonas, Manaus, Amazonas, Brazil, 2 Fundação de Medicina Tropical Doutor Heitor Vieira Dourado, Manaus, Amazonas, Brazil, 3 Programa de Pós-Graduação em Imunologia Básica e Aplicada, Instituto de Ciências Biológicas, Universidade Federal do Amazonas, Manaus, Amazonas, Brazil, 4 Faculdade de Medicina Nilton Lins, Universidade Nilton Lins, Manaus, Amazonas, Brazil, 5 Faculdade de Ciências Farmacêuticas, Universidade Federal do Amazonas, Manaus, Amazonas, Brazil, 6 Genomic Health Surveillance Network: Optimization of Assistance and Research in The State of Amazonas–REGESAM, Manaus, Amazonas, Brazil

* ramasawm@gmail.com

## Abstract

*Leishmania* are intracellular protozoan parasites that cause a wide spectrum of clinical manifestations in genetically susceptible individuals with an insufficient or balanced Th1 immune response to eliminate the parasite. MiRNAs play important regulatory role in numerous biological processes including essential cellular functions. miR146-a acts as an inhibitor of interleukin 1 receptor associated kinase 1 (IRAK1) and tumour necrosis factor (TNF) receptor associated factor 6 (TRAF6) present in the toll-like receptors pathway while miR499a modulates TGF-β and TNF signalling pathways. Here, we investigated whether *MIRNA146A rs2910164 and MIRNA499 rs3746444* variants are associated with the development of *L. guyanensis* (*Lg*)-cutaneous leishmaniasis (CL). The variants *MIR146A* rs2910164 and *MIR499A* rs3746444 were assessed in 850 patients with *Lg*-CL and 891 healthy controls by polymerase chain reaction and restriction fragment length polymorphism (PCR-RFLP). Plasma cytokines were measured using the BioPlex assay. Carriers of rs2910164 CC genotype have 30% higher odds of developing CL (ORadj$_{age/sex}$ = 1.3 [95% CI 0.9–1.8]; Padj$_{age/sex}$ 0.14) compared to individuals with the genotype GG (ORadj$_{age/sex}$ = 0.77 [95%CI 0.56–1.0]; Padj$_{age/sex}$ 0.14) if exposed to *Lg*-infection. Heterozygous GC individuals also showed lower odds of developing CL (ORadj$_{age/sex}$ = 0.77 [95%CI 0.5–1.1]; Padj$_{age/sex}$ 0.09). Homozygosity for the allele C is suggestive of an association with the development of *Lg*-CL among exposed individuals to *Lg*-infection. However, the odds of developing CL associated with the CC genotype was evident only in male individuals

**Data Availability Statement:** All relevant data are within the manuscript and its Supporting Information files.

**Funding:** This research was funded by the Brazilian Council for Scientific and Technological Development (CNPq), grant number 404181/2012-0 to Rajendranath Ramasawmy, Fundação de Amparo e Pesquisa do Estado do Amazonas (FAPEAM), grant number 06201954/2015 to Rajendranath Ramasawmy and FAPEAM RESOLUÇÃO N. 006/2020 - POSGRAD 2020. The funders had no role in study design, data collection and analysis, decision to publish, or preparation of the manuscript.

**Competing interests:** The authors have declared that no competing interests exist.

(OR$_{adjage}$ = 1.3 [95% CI = 0.9–2.0]; P$_{adjage}$ = 0.06). Individuals homozygous for the G allele tend to have higher plasma IL-8 and CCL5. Similarly, for the *MIR499A* rs3746444, an association with the G allele was only observed among male individuals (OR = 1.4 [1.0–1.9]; P = 0.009). In a dominant model, individuals with the G allele (GG-GA) when compared to the AA genotype reveals that carriers of the G allele have 40% elevated odds of developing *Lg*-CL (ORadj$_{age}$ = 1.4 [1.1–1.9]). Individuals with the GG genotype have higher odds of developing *Lg*-CL (ORadj$_{age/sex}$ = 2.0 [95%CI 0.83–5.0]; P$_{adjage}$ = 0.01. Individuals homozygous for the G allele have higher plasma IL-8. Genetic combinations of both variants revealed that male individuals exposed to *Lg* bearing three or four susceptible alleles have higher odds of developing *Lg*-CL (OR = 2.3 [95% CI 1.0–4.7]; p = 0.017). Both *MIR146A* rs2910164 and *MIR499A* rs3746444 are associated with the development of *Lg*-CL and this association is prevalent in male individuals.

## Author summary

Leishmaniasis is caused by infection with *Leishmania* parasites. In regions with the presence of *Leishmania* parasites, all people do not develop the disease despite similar exposure. Only a proportion of inhabitants progress to the development of disease. Clinical manifestations depend on the vector and *Leishmania* species, as well the host genetic background and genetically determined immune responses. miRNAs play important roles in regulating gene expression and many biological processes including immune pathways. miR-146a targets *TRAF6* and *IRAK1* genes, that encode key adaptor molecules downstream of toll-like receptors (TLRs). TLRs are critical in immune response to *Leishmania*-infection. miR499-a modulates inflammation-related signalling pathways such as TGFβ, TNFα and TLR pathways. In this study, we showed that *MIR146A* and *MIR499A* variants are risk factors to developing cutaneous leishmaniasis caused by *L. guyanensis* in Amazonas state of Brazil. Individuals with these variants are susceptible to the development of CL.

## Introduction

Leishmaniasis, a vector-borne disease caused by protozoan parasites from the genus *Leishmania*, *is endemic in* the tropical and subtropical regions, including more than 98 countries. Nearly, one billion of people are at risk of infection [1]. *Leishmania* species cause a spectrum of clinical forms of the disease: visceral (VL), cutaneous (CL), diffuse cutaneous, disseminated cutaneous and mucocutaneous leishmaniasis (ML) [2].

CL is considered the most common form of *Leishmania*-infection. Approximately 0.7 million to 1.2 million human beings are affected by this disease [1,3]. In Brazil, the main species involved in the etiology of CL are *L. braziliensis (Lb)*, *L. guyanensis (Lg)*, *L. lainsoni*, *L. amazonensis*, *L. shawi*, *L. naiffi* and *L. lindbergi* [4]. In the Amazonas state of Brazil, *Lg* is responsible for 95% of CL cases [5].

Resistance to *Leishmania*-infection or healing is associated with a Th1 cell immune response and production of pro-inflammatory cytokines (IL-12, IFN-γ and TNF-α) [6]. Susceptibility to infection and uncontrolled parasitic replication are associated with a Th2 anti-

inflammatory cytokines IL-4, IL-5 and IL-13 [7]. Th17 cells, producing IL-22 and IL-17, and T regulatory (Treg) cells producing IL-10 and TGF-β, also contribute to disease progression [6].

MicroRNAs (miRNAs) are small, single-stranded, untranslated endogenous RNAs, composed of 18–26 nucleotides that regulate gene expression. miRNAs bind to their target messenger RNA (mRNA) through base complementarity mechanism resulting in the regulation or degradation of protein translation [8,9]. More than 30% of human protein-coding genes are under post-transcriptional control of miRNAs [10]. miRNAs can regulate many physiological processes such as cell cycle, metabolism, and apoptosis. miRNAs play a crucial role in haematopoiesis, immune cells development and differentiation, inflammation, and immunomodulation [11,12].

MiRNAs can modulate macrophages polarization during *Leishmania*-infection, creating mix M1/M2 as shown in murine macrophages infected with *L. amazonensis* [13,14]. *L. donovani* glycoprotein can downregulate pre-miRNA-122 affecting post-transcriptional regulation of host mRNA/miRNA interactions leading to accumulation of the parasites in the liver [15]. *L. amazonensis* upregulates miR-294-30 and miR-721 that bind to *Nos2* mRNA leading to low levels of NOS2 and NO and increased infectivity in BALB/c-BMDM [16]. miRNAs, let-7a, miR-25, miR-26a, miR-132, miR-140, miR-146a, and miR-155 are upregulated in *L. major*-infected human macrophages and negatively correlated with the expression of their corresponding chemokine targets, CCL2, CCL5, CXCL10, CXCL11, and CXCL12 [17]. let-7a, let-7b, and miR-103 are upregulated in *L. donovani*-infected DCs and macrophages but downregulated in *L. major* infections [9]. Recently, it was shown that a super-enhancer mediates the transcription of *MIR146A* and drives the polarization of macrophages into M2 suppressing immune responses in *L. donovani*-infection [18].

miR146-a is a key modulator of innate immune response and acts as an inhibitor of interleukin 1 receptor associated kinase 1 (IRAK1) and tumour necrosis factor (TNF) receptor associated factor 6 (TRAF6) [19]. Both IRAK1 and TRAF6 are signalling transducers of the nuclear factor kappa B (NF-κB) in the Toll-like receptors (TLR) pathways. Inhibitions the NF-κB transcriptional activity leads to the impairment of the biosynthesis of pro-inflammatory cytokines such as IL-1β, IL-6, IL-8, and TNF-α [19]. miR499-a modulates different biological functions, including immune cells development and maturation, TGF-β and TNF signalling pathways [20]. miR499-a is correlated to the expression of IL-17RB, IL-23a, IL-2R, IL-6, IL-2, and IL-18R [21].

Single nucleotide polymorphisms (SNPs) in miRNA precursors may affect the miRNA biogenesis, causing a reduction of mature miRNA expression levels [22]. SNPs in mature miRNA may affect miRNA target specificity and leads to altered cellular protein levels [23,24]. These SNPs or variants in miRNA may by their actions alter the course of various diseases.

*MIR499A* and *MIR146A* genes are located on 20q11.22 and 5q33.3 chromosomes, respectively. The variant rs3746444 located in the seed region of mature miR499-a leads to the disruption of miRNA-mRNA interactions and creation of new gene targets [20]. The variant rs2910164 present in the seed sequence of miR146-a precursor results in low production of mature miR146-a and consequently to a decrease inhibition of TRAF6 and IRAK1, leading to a higher production of pro-inflammatory cytokines upon TLRs activation [25]. miRNA146-a and miR499-a have been associated with susceptibility to multiple types of cancer, psoriasis and rheumatoid arthritis [26–29].

Taken into account the potential role of miR146-a and miR499-a in modulating the T helper cell immune response, we investigated whether the variants *MIR146A* rs2910164 and *MIR499A* rs3746444 are associated with susceptibility or protection to the development of *Lg*-CL in the Amazonas. The influence of the *MIR146A* and *MIR499A* genotypes on plasma cytokine levels were also investigated.

## Methods

### Area of study and population

The study was conducted at the Fundação de Medicina Tropical Doutor Heitor Vieira Dourado (FMT-HVD), the referral regional center for treatment of leishmaniasis. The study population and the endemic area of recruitment of the participant of the study are described elsewhere [30]. Briefly, all the participants are from the perirural areas of Manaus, the capital city of the Amazonas State where *L. guyanensis* is endemic. Patients with active CL were followed at the FMT-HVD. The healthy controls with no history of CL and devoid of any scar suggestive of CL are from the same endemic area of the patients with CL.

### Ethical statement

This study was conducted according to the Declaration of Helsinki and approved by the Research Ethics Committee of the FMT-HVD granted under the file number CAAE:09995212.0.0000.0005 on 31 May 2013. All the participants or their responsible party for individuals less than 18 years of age provided written informed consent for the collection of samples and subsequent analysis.

### Sample size calculation

Sample size calculation is described elsewhere [31]. Briefly, using the Genetic Power calculator developed at Harvard University (http://pngu.mgh.harvard.edu/~purcell/gpe), we assumed a minor allele frequency of 5%, disease prevalence of 5%, complete linkage disequilibrium 1 between marker and case-control discrete trait, case-control ratio 1, and 5% type 1 error rates with an odds ratio of 1.5 and 2.0 for heterozygosity and homozygosity, respectively. For 80% power, the genetic allelic model provided a sample size of 789 individuals for cases and 789 for controls.

### *Leishmania spp* identification from biopsy specimens and DNA extraction from whole blood for SNP typing

All the patients with CL provided a biopsy specimen from the cutaneous lesion for the isolation of parasite DNA. For the identification of the *Leishmania spp*., the discrimination of the *Leishmania Viannia* subgenus specific PCR was in accordance with established protocols [32,33]. Identification of *Leishmania spp*. was performed by nucleotide sequencing of a fragment of *HSP 70* and *miniexon* genes [34,35]. Venous blood was drawn from all participants and collected into EDTA-containing Vacutainers (Becton Dickinson, Brazil) to DNA extraction and cytokine assay. Genomic DNA was extracted by the salting out method [36].

### SNP genotyping

The SNPs *MIR146A* rs2910164 and *MIR499A* rs3746444 were performed by PCR-RFLP with the restriction enzymes *HpyCH4 III* and *HpyCH4 IV* (New England Biolabs, Ipswich, MA, United States), respectively. The respective pairs of primers for the amplification the region flanking the SNPs and the fragments generated for alleles discrimination by the restriction enzymes as well as the cycling protocols for PCR are shown in S1 Table.

The pair of primers for *MIR499A* was designed from the reference sequence NC 000020.11 from NCBI. The underline G nucleotide from the forward primer substitutes A from the reference sequence to eliminate a site of restriction for the HpyCH4IV. Primers for the *MIR146A*

were designed from the reference sequence NC 000005.10 and the underline A nucleotide substitutes C from the reference sequence to create restriction site when the G allele is present.

The PCR mix contains: 0.2 µM of each primer (Thermofisher, MA USA), 40 nM of dNTP (Thermofisher, MA USA), 1.0 mM of MgCl2 (Thermofisher, MA USA), 1 U of Taq DNA polymerase (Thermofisher, MA USA), 1X of 10X Taq polymerase buffer containing 500 mmol/L KCl and 100 mol/L of Tris-HCL (pH 8.3) and 50 ng of DNA in a final volume of 25 µL. A volume of 10 µL of the PCR products was digested with 1 unit of the respective restriction enzyme and buffer in a final volume of 20 uL and size separated in a 3% agarose (Ultrapure Agarose, Thermofisher, MA USA) gel electrophoresis.

## Cytokine assay

Cytokine assay of IL-1β, TNF-α, IL-2, IL-6, IL-8, IL-17, IFN-γ, CCL2 and CCL5 in the plasma were measured using the multiplex cytokine commercial kit Human Cytokine Grp I Panel 27-Plex (Bio-Rad, USA) according to the instructions of the manufacturer in the Bio-plex 200 Protein Array System (Luminex Corporation, USA).

## Statistical analysis

The genotype and allele frequencies were calculated by direct gene counting. For calculation of Hardy-Weinberg equilibrium (HWE), the website http://ihg.gsf.de/cgibin/hw/hwa1 was used, that also compared cases with the control groups by logistic regressions analysis to determine associations to susceptibility or resistance for the different genotypes and alleles by χ2 test along with OR and 95% confidence interval. The correlation of the different genotypes of *MIR146A* rs2910164 and *MIR499A* rs3746444 to the concentration of circulating plasma cytokines was performed by the R software version 4.0.0 of SNPassoc package for quantitative traits analysis and ggplot2 package for visualizing. P values for the correlations of cytokines by genotypes were corrected for false discovery rate (FDR) of Benjamini-Hochberg.

# Results

## Study population

The study population is the same as described previously [30]. A total of 850 patients with *Lg*-CL and 891 healthy controls (HC) were included in the study. The HC are from the same endemic area of the patients with *Lg*-CL. Among the patients with *Lg*-CL, 639 (75%) patients were male (mean age 34.4 ± SD 13.7 years) and 211 (25%) were females (37.5± SD 15.7 years). In the controls group, 608 (68%) were male (42± SD 17.5 years) and 283 (32%) were female (40± SD 17.4 years). Overall, the mean age among the patients with *Lg*-CL and controls is 35.17± SD 14.25 and 41.4± SD 17.5 years, respectively. The HC is older than the cases (P<0.0001). Men were older among the HC group compared to group of male patients with *Lg*-CL (P<0.0001) while there was no age difference among females (P<0.077). All the participants of the study were devoid of HIV and the patients had fewer or equal to six lesions and treatment naïve at the time of enrolment. Pregnant women were excluded from the study. Of note, there are more females in the HC compared to the group of patients with CL (P<0.0013).

## Association *MIR146A* rs2910164 and *MIR499*A rs3746444 with susceptibility to Cutaneous Leishmaniasis

*MIR146A* rs2910164 was assessed in 826 patients with *Lg*-CL and 886 controls. Genotype and allele frequencies for the two variants are demonstrated in Table 1. rs2910164 was in Hardy-

**Table 1. Genotype and allele frequencies for the *MIR146A* rs2910164 and *MIR499A* rs3746444 in patients with *Leishmania guyanensis*-Cutaneous Leishmaniasis (*Lg*-CL) and healthy controls.**

| Genotypes and Alleles Frequencies | | | | |
|---|---|---|---|---|
| | Cases[a] | (%) | HC[b] | (%) |
| rs2910164 | N = 826 | | N = 886 | |
| GG | 375 | 46 | 405 | 46 |
| GC | 349 | 42 | 398 | 45 |
| CC | 102 | 12 | 83 | 9 |
| G | 1099 | 67 | 1209 | 68 |
| C | 553 | 33 | 563 | 32 |
| rs3746444 | N = 818 | | N = 851 | |
| AA | 649 | 79 | 706 | 83 |
| AG | 153 | 19 | 135 | 16 |
| GG | 16 | 2 | 10 | 1 |
| A | 1451 | 89 | 1547 | 91 |
| G | 185 | 11 | 155 | 9 |
| Genotypes and alleles comparisons | | | | |
| Comparisons | P value[c] | OR [95% CI][d] | Padj[e] | ORadj[95%CI][f] |
| rs2910164 | | | | |
| GG vs CC | 0.084 | 1.3 (0.96–1.8) | 0.14 | 1.3 (0.9–1.8) |
| CC vs GC | 0.040 | 1.4 (1.0–1.9) | 0.09 | 1.3 (0.9–1.8) |
| CC+GC vs GG | 0.897 | 1.0 (0.8–1.2) | 0.864 | 1.0 (0.8–1.2) |
| GG+GC vs CC | 0.047 | 1.4 (1.0–1.9) | 0.098 | 1.3 (0.9–1.8) |
| G vs C | 0.304 | 0.9 (0.8–1.0) | | |
| rs3746444 | | | | |
| AA vs GG | 0.167 | 1.7 (0.8–3.9) | 0.15 | 2.0 (0.83–5.0) |
| AA vs AG | 0.106 | 1.2 (0.9–1.6) | 0.13 | 1.3 (1.0–1.7) |
| AA+GA vs GG | 0.196 | 1.7 (0.8–3.7) | 0.170 | 1.8 (0.8–4.0) |
| AA vs GA+GG | 0.058 | 1.3 (1.0–1.6) | 0.067 | 1.3 (1.0–1.6) |
| A vs G | 0.035 | 1.3 (1.0–1.6) | | |

[a]Cases: patients with *Lg*-CL

[b]HC: healthy controls

[c]*P* value: normal *p*-value

[d]OR: odds ratio with 95% confidence interval (CI)

[e]*P*adj: p adjusted by gender and age

[f]ORadj: odds ratio adjusted by gender and age; *P* value <0.05 is statistically significant.

Weinberg equilibrium (HWE) among the patients with *Lg*-CL and HC. The *MIR146A* rs2910164 CC genotype was prevalent in patients with *Lg*-CL (12%) compared with HC (9%). Carriers of rs2910164 CC genotype have 30% higher odds of developing CL (ORadj$_{age/sex}$ = 1.3 [95%CI 0.9–1.8]; Padj$_{age/sex}$ = 0.14) compared to individuals with the genotype GG (ORadj$_{age/sex}$ = 0.77 [95%CI 0.56–1.0]; Padj$_{age/sex}$ = 0.14) if exposed to *Lg*-infection. Heterozygous GC individuals also have lower odds of developing CL compared with homozygous carriers of the C allele (ORadj$_{age/sex}$ = 0.77 [95%CI 0.55–1.1]; Padj$_{age/sex}$ = 0.09). In a recessive model, when homozygous individuals for the C allele are compared with individuals carrying a G allele (CC versus GC—GG), carrier of G allele have 23% lower odds of developing *Lg*-CL (ORadj$_{age/sex}$ = 0.77 [95%CI 0.56–1.0]; Padj$_{age/sex}$ = 0.098). Homozygous for the C allele have 30% higher odds of developing CL (ORadj$_{age/sex}$ = 1.3 [95%CI 0.9–1.8]; Padj$_{age/sex}$ = 0.098). The C allele is suggestive of an association with the development of *Lg*-CL.

**Table 2. Distribution and Comparison of Genotypes and Alleles frequencies of the *MIR146A* rs2910164 between Patients with *Leishmania guyanensis*-Cutaneous Leishmaniasis (*Lg*-CL) and Controls stratified according to gender.**

| | Patients with *Lg*-CL, no. (%) | | | Controls, no. (%) | |
|---|---|---|---|---|---|
| Genotypes | Males | Females | | Males | Females |
| | N = 621 | N = 205 | | N = 603 | N = 283 |
| G/G | 284 (46) | 91 (44) | | 286 (47) | 120 (42) |
| G/C | 254 (41) | 95 (47) | | 261 (44) | 136 (48) |
| C/C | 83 (13) | 19 (9) | | 56 (9) | 27 (10) |
| Alleles | | | | | |
| G | 822 (66) | 277 (67) | | 833 (69) | 376 (66) |
| C | 420 (34) | 133 (33) | | 373 (31) | 190 (34) |

| Genotypes and alleles comparisons | | | | | | | | |
|---|---|---|---|---|---|---|---|---|
| | Males | | | | Females | | | |
| | P v[a] | OR[b] [95%CI] | Padj[c] | ORadj[d] [95%CI] | Pv[a] | OR[b] [95%CI] | Padj[c] | ORadj[d] [95%CI] |
| GG vs CC | 0.036 | 1.5 [1.0–2.2] | 0.06 | 1.3 [0.9–2.0] | 0.82 | 0.9 [0.5–1.8] | 0.45 | 1.0 [0.5–1.8] |
| CC vs GC | 0.03 | 1.5 [1.0–2.2] | 0.04 | 1.4 [1.0–2.0] | 0.98 | 1.0 [0.5–1.9] | 0.45 | 1.0 [0.5–2.0] |
| GG+GC vs CC | 0.02 | 1.5 [1.0–2.2] | 0.04 | 1.4 [0.95–2.0] | 0.92 | 1.0 [0.5–1.9] | 0.38 | 1.0 [0.7–1.5] |
| G vs C | 0.12 | 1.1 [0.96–1.3] | | | 0.71 | 1.1 [0.8–1.4] | | |

[a]*P* v: normal *p*-value

[b]OR: odds ratio with 95% confidence interval (CI)

[c]*P*adj: p adjusted by age

[d]ORadj: odds ratio adjusted by age. *P* value < 0.05 is statistically significant.

As male individuals were prevalent in both patients with *Lg*-CL and HC, we stratified according to sex as shown in Table 2. The odds of developing CL associated with the CC genotype was evident only in male individuals (OR$_{adjage}$ = 1.3 [95% CI = 0.9–2.0 P$_{adjage}$ = 0.06]) compared with GG genotype. Individuals with GG genotype had lower odds of developing *Lg*-CL (R$_{adjage}$ = 0.77 [95%CI 0.5–1.0]; p = 0.06). The genotype CC is suggestive of an association with the development of CL compared to GG genotype among *Lg*-infected individuals.

*MIR499A* rs3746444 was genotyped in 818 patients with *Lg*-CL and 851 HC. The variant was in HWE in both groups of patients with *Lg*-CL and HC. The distribution of genotypes among patients with *Lg*-CL and HC was different as shown in Table 1, revealing a common odds ratio of 1.3 for the G allele compared with the A allele (p = 0.04). The rs3746444 AA genotype was more prevalent among the HC (83%) compared to 79% in the patients with *Lg*-CL group. Comparison of the genotype AA to GG revealed that individuals with the GG genotype have 100% higher odds of developing *Lg*-CL, with a 95% CI ranging from a decreased odds of 17% to an elevated odd of 400% (ORadj$_{age/sex}$ = 2.0 [95%CI 0.83–5.0]; Padj$_{age/sex}$ = 0.15). Similarly, the G allele confers 27% elevated odds of developing *Lg*-CL suggesting that the G allele contribute to susceptibility to the development of *Lg*-CL (OR = 1.27 (95% CI 1.0–1.6]; p = 0.035). In a dominant model, individuals with the G allele (GG-GA) when compared to the AA genotype reveals that carriers of the G allele have 30% elevated odds of developing *Lg*-CL (ORadj$_{age/sex}$ = 1.3 [95% CI 1.0–1.6]; Padj$_{age/sex}$ = 0.067).

We stratified into male and female individuals to look for the strength of the association. Like the *MIR146A*, the association was more evident among the male individuals as shown in Table 3. Male individuals with GG genotype compared with AA genotype revealed an elevated odds of 50% with a 95% confidence interval ranging from a decreased odds of 40% to an increased odds of 310% to the development of *Lg*-CL (OR$_{adj/age}$ = 1.5 [0.6–4.1]; P$_{adj/age}$ = 0.38).

**Table 3. Distribution and Comparison of Genotypes and Alleles frequencies of the *MIR499A* rs3746444 between Patients with *Leishmania guyanensis*-Cutaneous Leishmaniasis (*Lg*-CL) and Controls stratified according to gender.**

| Genotypes | Patients with *Lg*-CL, no. (%) | | Controls, no. (%) | |
|---|---|---|---|---|
| | Males | Females | Males | Females |
| | N = 614 | N = 204 | N = 538 | N = 268 |
| A/A | 478 (78) | 171 (84) | 488 (84) | 218 (81) |
| A/G | 125 (20) | 28 (14) | 88 (15) | 47 (18) |
| G/G | 11 (2) | 5 (2) | 7 (1) | 3 (1) |
| Alleles | | | | |
| A | 1081 (88) | 370 (91) | 1064 (91) | 483 (90) |
| G | 147 (12) | 38 (9) | 102 (9) | 53 (10) |

| | Genotypes and alleles comparisons | | | | | | | |
|---|---|---|---|---|---|---|---|---|
| | Males | | | | Females | | | |
| | $P$ v[a] | OR[b] [95%CI] | $P$adj[c] | ORadj[d] [95%CI] | $P$ v[a] | OR[b] [95%CI] | $P$adj[c] | ORadj[d] [95%CI] |
| AA vs. GG | 0.32 | 1.6 [0.6–4.1] | 0.38 | 1.5 [0.6–4.1] | 0.29 | 2.1 [0.5–8.0] | 0.26 | 2.3 [0.5–10.0] |
| AA vs. AG | 0.01 | 1.4 [1.0–1.9] | 0.02 | 1.4 [1.0–1.9] | 0.28 | 1.3 [0.8–2.2] | 0.29 | 1.3 [0.8–2.2] |
| AA vs AG+GG | 0.01 | 1.5 [1.0–1.9] | 0.01 | 1.4 [1.1–1.9] | 0.48 | 1.2 [0.7–1.9] | 0.51 | 1.2 [0.7–1.9] |
| A vs. G | 0.009 | 1.4 [1.0–1.9] | | | 0.76 | 1.1 [0.7–1.7] | | |

[a]$P$ value: normal *p*-value

[b]OR: odds ratio with 95% confidence interval (CI)

[c]$P$adj: p adjusted by age

[d]ORadj: odds ratio adjusted by age. *P* value < 0.05 is statistically significant.

Similarly, female individuals had higher odds of developing *Lg*-CL (OR$_{adj}$ = 2.3 [95%CI 0.5–10.0]; Padj$_{age/sex}$ = 0.26).

## Genetic combinations of both *MIR146A* rs2910164 and *MIR499A* rs3746444 genotypes

The frequencies of the combined genotypes are shown in Supplementary Table 3. Male individuals carrying at least one susceptibility allele (*MIR146A* rs2910164C or *MIR499A* rs3746444G) revealed an OR of 1.1 [95%CI 0.8–1.4] in comparison to individuals with two alleles (1.4 [95% CI 1.0–1.9]; p = 0.035) or three or four alleles (2.3 [95% CI 1.0–4.7]; p = 0.017). Notably, these associations were not observed among female individuals strengthening that the associations are mainly among male individuals.

## Comparison of circulating plasma cytokines concentrations (pg/mL) with respect to the *MIR146A* rs2910164 and *MIR499A* rs3746444 genotypes

miRNA146-a is suggested to suppress IRAK-1 expression leading to a decrease of IL-6 and IL-8 secretions that are key mediators of inflammation [37]. Bone-marrow derived macrophages from knockout *MIR146A*$^{-/-}$ mice treated with monosodium urate expressed higher levels of IL-1β, TNFα, NLRP3, IRAK-1 and TRAF-6 compared with BMDM from wild type mice [38]. Inhibitors of miR146-a stimulate the expression of IL-8 and CCL5 [39]. We analysed whether the different genotypes of *MIR146A* may correlate with circulating plasma levels of IL-6, IL-8, IL-1β, TNFα and CCL5 (S1 Fig).

Only IL-8 showed a tendency to correlate with the genotypes of rs2910164 in healthy individuals albeit no statistical significance is reached. In a dominant model, individuals homozygous for the G allele seem to have higher levels of circulating IL-8 compared with individuals

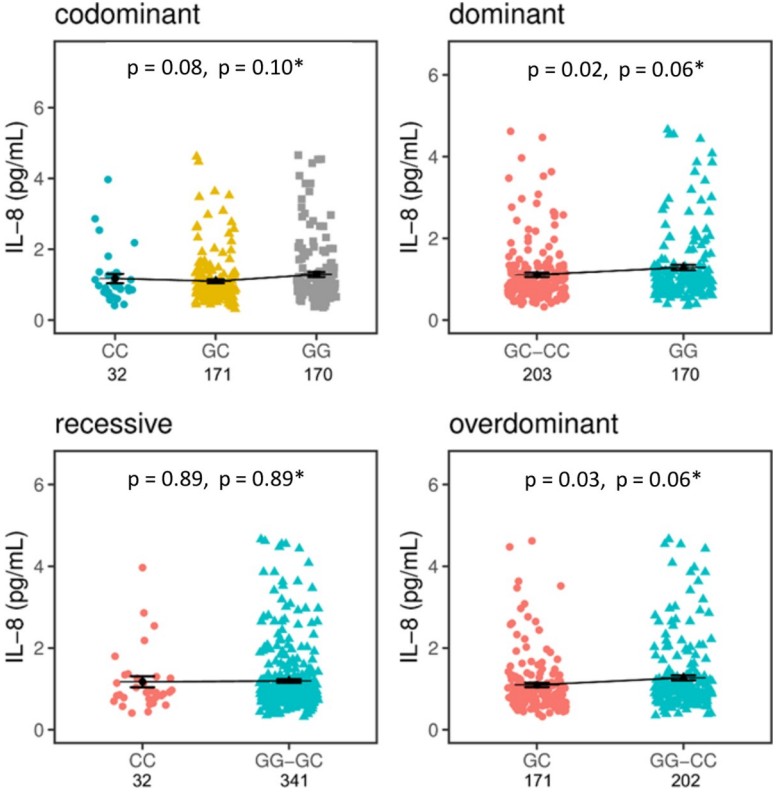

**Fig 1. *MIRNA146A* rs2910164 effects on IL-8 plasma levels in healthy controls.** Concentrations of plasma circulating IL-8 were compared between bearers of the AA, AG, and GG genotypes using inheritance models (codominant, dominant, recessive and overdominant). The mean concentrations are displayed in picogram per millilitre (pg/mL) with standard error (SE) of mean. The means are represented by black bars, whereas SE are represented by error bars. *Corrected for Benjamini-Hochberg False Discovery Rate assuming four tests. p < 0.05 is statistically significant.

bearing a C allele (GG 1.28±0.06 pg/mL vs CC +GC 1.10±0.05 pg/mL; p = 0.02; P corrected for false discovery rates ($P_{FDR}$) = 0.06) as shown in Fig 1. Similarly, these individuals seem to have increased levels of CCL5 (GG 73.01±30.14 pg/mL compared with individuals with a C allele, CC +GC 31.68±2.68 pg/mL; p = 0.19) (S2 Fig).

miRNA499-a modulates many inflammation-related signalling pathways such as TGFβ, TNFα and Toll-like receptors pathways. miR499-a has also been reported to regulate the expression of TNFα, IL-6, IL17RB, IL-18R and IL-23a [21]. We assayed IL-1β, IL-2, IL-6, IL-8, IL-17, IFNγ, CCL2 and TNFα to correlate with the different genotypes of *MIR499A* rs374644 as shown in S3 Fig. A tendency of plasma circulating levels of IL-8, IL-6 and IL-17 by *MIR499A* rs3746444 genotypes is observed. Among the HC, the distribution of plasma circulating IL-8 according to the genotypes is statistically different in a codominant model (p = 0.01; $P_{FDR}$ = 0.02) as shown in Fig 2. In a recessive model, individuals homozygous for the G allele seem to have higher levels of IL-8 (2.14±0.64 pg/mL) compared with individuals (AA +GA 1.18±0.04 pg/mL) bearing a A allele (p = 0.003; $P_{FDR}$ = 0.01) (Fig 2).

GG homozygotes have higher levels of IL-8 (GG = 2.14±0.64 pg/mL versus AA = 1.18 ±0.04pg/mL (p = 0.005; $P_{FDR}$ = 0.01) (Fig 3).

In a recessive model, bearers of the A allele among HC (AA+GA; 0.40±0.01 pg/mL) seem to have lower levels of IL-6 (p = 0.09) compared with GG homozygotes (0.64±0.18 pg/mL) (**S4 Fig**). Individuals homozygous for the G allele have higher levels of IL-6 compared with AA

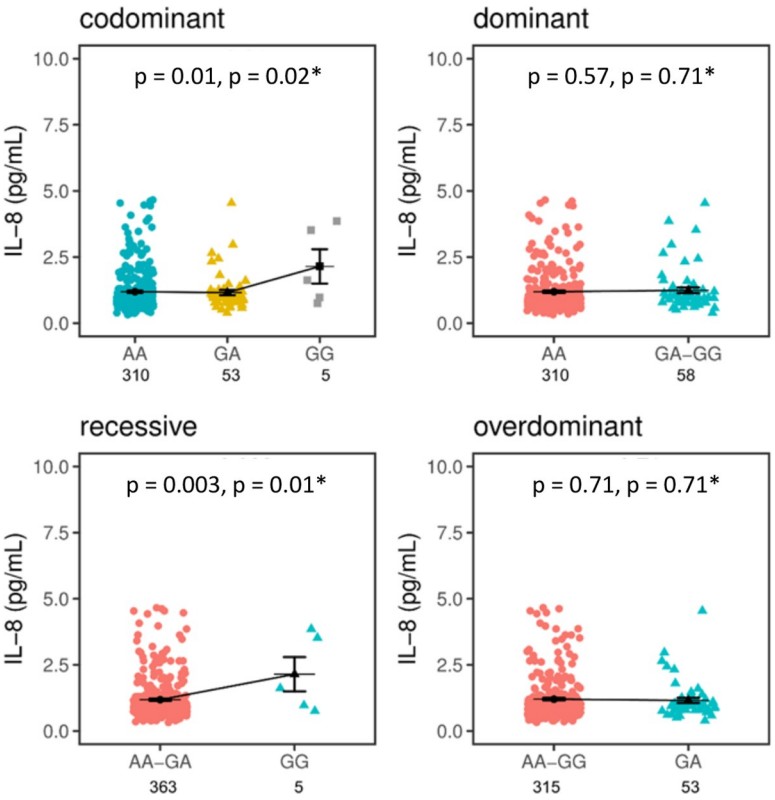

**Fig 2. *MIRNA499A* rs3746444 effects on IL-8 plasma levels in healthy controls.** Concentrations of plasma circulating IL-8 were compared between bearers of the AA, AG, and GG genotypes using inheritance models (codominant, dominant, recessive and overdominant). The mean concentrations are displayed in picogram by millilitre (pg/mL) with standard error (SE) of mean. The means are represented by black bars, whereas SE are represented by error bars. *Corrected for Benjamini-Hochberg False Discovery Rate assuming four tests. $p < 0.05$ is statistically significant.

homozygotes individuals (GG = 0.64±0.18 pg/mL versus AA = 0.41±0.01 pg/mL; p = 0.09) (Fig 3).

Similarly, *MIR499A* rs3746444 have a tendency on influencing the plasma circulating levels of IL-17 among the HC. GG homozygotes (5.48±1.97 pg/mL) have higher levels of IL-17

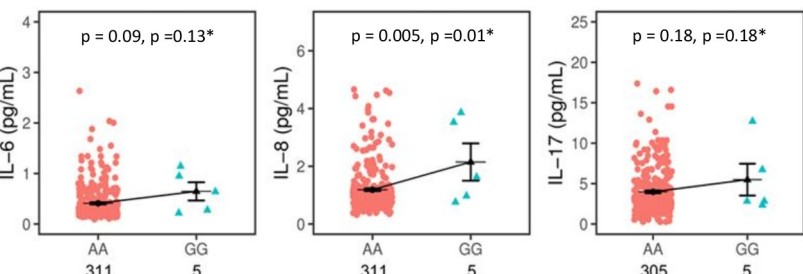

**Fig 3. *MIRNA499A* rs3746444 effects on IL-6, IL-8, and IL-17 plasma levels between AA homozygous versus GG homozygous in healthy controls.** Comparisons of circulating plasma levels of IL-6, IL-8 and IL-17 between the individuals AA and GG homozygous. The mean concentrations are displayed in picogram per millilitre (pg/mL) with standard error (SE) of mean. The means are represented by black bars, whereas SE are represented by error bars. *Corrected for Benjamini-Hochberg False Discovery Rate assuming three tests. $p < 0.05$ is statistically significant.

compared to heterozygotes GA (3.48±0.31 pg/mL) and homozygotes AA (3.98±0.16 pg/mL) (S4 Fig).

Genetic combinations of both variants revealed that individuals bearing three or four G alleles tend to correlate with higher plasma IL-8 (2.24 ± 0.48 pg/mL) and CCL5 (63.4± 29.2 pg/mL) as shown in S5 Fig.

## Discussion

The manifestation of clinical symptoms in *Leishmania*-infected individuals depends on the capacity of the individual to have a fine regulation of the TH1 response to eliminate the parasite and avoid an exacerbation of expression of pro-inflammatory cytokines. Emerging evidence are showing that miRNAs regulate immune response [19,40,41].

In this study, we showed that male individuals homozygous for the C allele of *MIR146A* rs2910164 have 30% higher odds of developing *Lg*-CL compared with individuals bearing a G allele. The *MIR146A* rs2910164 C allele has been shown to be a risk factor for leprosy in Brazil and psoriasis in South African Indian [42,43]. Homozygous individuals for the C allele have increased risk of developing glioma and decreased survival [44]. A meta-analysis study on autoimmune diseases revealed that individuals bearing the C allele have increased risk of developing the disease [45].

Several studies have also reported that *MIR146A* rs2910164 is not a risk factor in several diseases. A meta-analysis on ischemic stroke showed that *MIR146A* rs2910164 is not associated with any risk occurrence of ischemic stroke [46]. A lack of association of *MIR146A* rs2910164 with the development of RA was also reported [47]. Another study cited that susceptibility to pulmonary tuberculosis is not influenced by the *MIR146A* rs2910164 [48]. However, a meta-analysis approach on psoriasis case-control studies with rs2910164 revealed that the CC genotype is correlated with decreased risk of psoriasis [28].

Conversely, other studies have reported that the G allele is a risk factor in several diseases. The G allele has been associated with pulmonary TB [49] and ankylosing spondylitis [50]. A meta-analysis study on cancer among Asian patients revealed that GG genotypes are associated with increased risk of cancer [51].

We also showed that in male individuals, the GG genotype of the *MIR499A* rs3746444 is associated with 50% increased odds of developing CL. The G allele confers 27% elevated odds of developing CL suggesting that the G allele may contribute to the development of *Lg*-CL.

A recent meta-analysis study showed that the *MIR499A* rs3746444 G allele is associated with high risk of developing breast cancer [52]. The allele G was also associated with hepatocellular carcinoma [53], and adenocarcinoma [54]. Other studies also showed that the G allele is associated with susceptibility to the development of bronchial asthma [20], RA [29,55], Behcet's disease [56], ankylosing spondylitis [50], myocardial infarction [57] and coronary artery disease [58,59]. Immune response plays critical role in all these diseases. Of note, there is no study of both variants with protozoan infectious diseases to date.

The gender difference observed for both variants in this study may be due to either sexual hormonal interaction with the variant or the small sample size in the female group. Notably, we have 75% and 68% male individuals among patients with CL and HC, respectively. Interestingly, sex-based differences have been shown in *L. tropica* infected patients. *L. tropica*-infection manifests commonly in the form of CL. However, females present predominantly CL while male individuals are more incline to develop viscerotropic leishmaniasis [60–62]. Furthermore, peripheral blood from patients with CL caused by *L. mexicana* stimulated or not with lipophosphoglycan demonstrated higher expression of IFNγ and tumour necrosis factor alpha in females than in males [63]. An increased parasite growth has been observed when *L.*

*mexicana* promastigotes are treated with physiological doses of male dihydrotestosterone [64,65].

The *MIR146A* is in the cytokine cluster 5q31 and the *MIR146A* rs2910164 is situated in the stem region of the *pre-miR-146a*. The change of guanine:uracil pair (G:U) to cytosine:uracil (C:U) in the stem structure affects the production of mature miR146-a [25,66]. The G-allele is reported to be associated with high expression levels of mature miR146-a while the C-allele with low expression [25, 67]. Other study cited that the C allele is correlated with higher miR146-a and lower TNFα expression from nerve biopsies of leprosy patients [42]. High expression of miR146-a was also observed in specimens of healthy and tumour tissue from patients with gastric cancer bearing genotype CC compared to GG [68]. In Lupus, CC genotypes was correlated with high miR146-a expression [69]. One study observed that the genotype CC is associated with increased expression of IRAK1 and TRAF6 [70]. miR146-a regulates the IL-17 pathway in human keratinocytes, the first line of defence against pathogens in the skin, to restrain IL17-induced inflammation [71]. Overexpression of miR146-a in keratinocytes inhibits the expression of IL-8 and TNFα, suppressing the chemotactic attraction of neutrophils by keratinocytes [72]. Interestingly, we observed in this study that individuals homozygous for the C allele have lower plasma IL-8. Generation of miRNA regulatory network from Cytoscape revealed that miR146-a play key role in the inflammatory response in *Leishmania*-infection [73].

In this study, we did not observe any correlation of plasma levels of TNFα, IL-6, IL-1β, IL-2, IL-17, IFNγ and CCL2 by *MIR146A* rs2910164 genotypes neither among the patients with *Lg*-CL nor the HC. However, individuals homozygous for the G allele seem to have higher levels of circulating IL-8 and CCL5 compared with individuals bearing a C allele.

Knockout *MIR146A*$^{-/-}$ mice developed exaggerated pro-inflammatory response upon challenging with lipopolysaccharide due to chronic dysregulation of nuclear factor kappa-light-chain enhancer of activated B cells (NFkB) signalling [74,75]. miR146-a is reported to target IRAK and TRAF-6 in the TLRs downstream pathway to downregulate the upregulation of NFkB leading to a decrease in the transcription of pro-inflammatory cytokines IL-6, IL-8, IL-1β, and TNFα [74,76–80]. A recent study showed that inhibition of miR146-a resulted in a reduced secretion of IL-6 and IL-8 [81], suggesting that miR146-a may downregulate an inflammatory reaction.

miRNA499-a is involved in TLR-signalling [82]. Computational tool analysis suggests that miR499-a may target IL-13 and IL-23 (microRNA.org). MIR*499A* rs3746444 genotype GG has been suggested to correlate with high expression of miR499-a [83]. Our data showed that individuals homozygous for the G allele have higher plasma IL-8 and a tendency to also have high IL-6, IL-17, and increased risk of developing *Lg*-CL.

Indeed, in *Lb*-infected patients with CL, high levels of IL-17 were observed [84] and peripheral blood from these patients released high IL-17 upon stimulation with soluble *Leishmania* antigen [85]. *L. major*-infected C57BL/6 mice developed larger lesion size with increased production of IL-17 and neutrophil infiltration at the site of lesion compared with mice treated with anti-IL-17 [85] while *L. major*-infected BALB/c mice deficient for IL-17 develop smaller lesions [86], suggesting that IL-17 promotes lesions. High levels of IL-6 impaired the cytokine-enhanced antileishmanial activity of human macrophages by inhibiting IFNγ and TNFα [87].

Our study has several limitations. Firstly, we considered that our healthy controls from the same endemic areas as the patients are exposed to the *Lg*-vector and are infected but did not develop the disease. We did not perform DTH to ensure this despite most of the individuals included in the study are farming or agricultural workers. Secondly, the problem of multiple testing of many associations may result in spurious associations but may also discard a true association after correcting for multiple comparisons. Thirdly, our controls are slightly older

than our patients with CL and contain slightly more females. Stratification reduces the sample size.

Considering our data, we may hypothesize that *MIR146A* rs2910164 CC individuals are prone to develop *Lg*-CL due to an impairment neutrophil migration at the inoculation of the parasite by the sand fly bite. Additionally, these individuals have lower levels of CCL5. CCL5 has been shown to correlate with resistance to *L. major* infection in animal model [88]. Furthermore, our *MIR499A* rs3746444 data suggest that individuals with genotype GG are susceptible to develop *Lg*-CL by attracting too many neutrophils that may pass the parasite to macrophages leading to parasite persistence as they might be high producers of IL-8, IL-6 and IL-17.

Altogether, this is the first study to date to investigate *MIR146A* rs2910164 and *MIR499A* rs3746444 in protozoan infectious diseases. Both variants are associated with the development of *Lg*-CL male individuals exposed to *Lg*-infection and correlate with plasma IL-8.

## Supporting information

**S1 Fig. Plasma levels of cytokines and chemokines according to the different genotypes of *MIR146A* rs2910164 in cases, controls and total individuals.**
(TIFF)

**S2 Fig. *MIR146A* rs2910164 effects on circulating plasma CCL5 levels in healthy controls.**
(TIFF)

**S3 Fig. Plasma levels of cytokines and chemokines according to the different genotypes of *MIR499A* rs3746444 in cases, controls and total individuals.**
(TIFF)

**S4 Fig. *MIR499A* rs3746444 effects on circulating plasma IL-6 and IL-17 levels in healthy controls.**
(TIFF)

**S5 Fig. Genetics combinations of genotypes by plasma cytokines.**
(TIFF)

**S1 Table. Polymerase Chain Reactions protocols for the *MIR146A* rs2910164 and *MIR499A* rs3746444.**
(DOCX)

**S2 Table. Mean values and standard error (SE) of the mean of plasma cytokines by the *MIR146A* rs2910164 and *MIR499A* rs3746444 genotypes according to inheritance models among the total individuals.**
(DOCX)

**S3 Table. Genetics combinations of genotypes in patients with Cutaneous Leishmaniasis and Healthy Controls.**
(DOCX)

## Acknowledgments

The authors would like to thank all patients and individuals from the endemic areas for participating in this work.

## Author Contributions

**Conceptualization:** Rajendranath Ramasawmy.

**Data curation:** Tirza Gabrielle Ramos de Mesquita, Lissianne Augusta Matos Gomes, Mara Lúcia Gomes de Souza, Rajendranath Ramasawmy.

**Formal analysis:** Tirza Gabrielle Ramos de Mesquita, José do Espírito Santo Junior, Mara Lúcia Gomes de Souza, Rajendranath Ramasawmy.

**Funding acquisition:** Rajendranath Ramasawmy.

**Methodology:** Tirza Gabrielle Ramos de Mesquita, José do Espírito Santo Junior, Thais Carneiro de Lacerda, Krys Layane Guimarães Duarte Queiroz, Cláudio Marcello da Silveira Júnior.

**Resources:** José Pereira de Moura Neto.

**Supervision:** Rajendranath Ramasawmy.

**Writing – original draft:** Tirza Gabrielle Ramos de Mesquita, Rajendranath Ramasawmy.

**Writing – review & editing:** Marcus Vinitius de Farias Guerra, Rajendranath Ramasawmy.

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
