## [Decision Letter · Decision Letter 0]

24 Jul 2021

Dear Dr. Ramasawmy,

Thank you very much for submitting your manuscript "Variants of MIRNA146A rs2910164 and MIRNA499 rs3746444 are associated with the development of cutaneous leishmaniasis caused by Leishmania guyanensis and with plasma chemokine IL-8" for consideration at PLOS Neglected Tropical Diseases. As with all papers reviewed by the journal, your manuscript was reviewed by members of the editorial board and by several independent reviewers. In light of the reviews (below this email), we would like to invite the resubmission of a significantly-revised version that takes into account the reviewers' comments. 

Please, check the comments of the reviewers. They raised important points that must be addressed to consider the manuscript for publication.

We cannot make any decision about publication until we have seen the revised manuscript and your response to the reviewers' comments. Your revised manuscript is also likely to be sent to reviewers for further evaluation.

Sincerely,

Claudia Ida Brodskyn

Associate Editor

Steven Singer

Deputy Editor

Please, check the comments of the reviewers. They raised important points that must be addressed to consider the manuscript for publication.

Reviewer's Responses to Questions

**Key Review Criteria Required for Acceptance?**

**Methods**

-Are the objectives of the study clearly articulated with a clear testable hypothesis stated?

-Is the study design appropriate to address the stated objectives?

-Is the population clearly described and appropriate for the hypothesis being tested?

-Is the sample size sufficient to ensure adequate power to address the hypothesis being tested?

-Were correct statistical analysis used to support conclusions?

-Are there concerns about ethical or regulatory requirements being met?

Reviewer #1: The objective of performing the polymorphism analysis is clear; the cytokine measurements are too indirect. The population is well described, although the sample is highly biased for sex. Some analysis accounted for that but not all. there is no information regarding sample size calculation. No concerns related to ethical requirements.

Reviewer #2: The methods is in consonante with goal proposed in the manuscript, supporting the objetives. The statistical analysis was adequate. 

A minor point can be included in the text:

What is the volume of PCR mix and Buffer? And the volume of enzymatic digestion mix?

Reviewer #3: Objectives are clearly stated and the study design and statistical analysis (except correction for multiple testing, see below) in agreement with these objectives. The authors should be recommended for an exceptionally large sample size, which is uncommon in neglected tropical diseases.

**Results**

-Does the analysis presented match the analysis plan?

-Are the results clearly and completely presented?

-Are the figures (Tables, Images) of sufficient quality for clarity?

Reviewer #1: Results are clearly presented.

Reviewer #2: Tables and Figures are clearly and showed that an analysis of SNPs MIR146A rs2910164 and MIR499A rs3746444 showed a prevalence of MIR146A rs2910164 CC genotype and MIR499A rs3746444 AA genotype, elevating the risk to develop CL. In MIR146A rs2910164 G allele presented higher levels of IL-8 and RANTES in CL plasma samples. The, MIR499A rs3746444 G allele presented higher levels of IL-8, IL-6 and IL-17.

This information open some questions:

- Did you analyze the prevalence of both MIR146A rs2910164 CC and MIR499A rs3746444 AA genotypes in the same CL patients? These correlates?

- In MIR146A rs2910164 G allele presented higher levels of IL-8 and RANTES in CL plasma samples. The, MIR499A rs3746444 G allele presented higher levels of IL-8, IL-6 and IL-17. Have some correlation of presence of both rs2910164 G allele and rs3746444 G allele and cytokines in the same CL patients?

Reviewer #3: Major comments:

1. Correction for multiple testing is strictly necessary for the cytokine analysis, since the Methods sections mentions 8 different cytokines/chemokines were measured: “Cytokine assay of IL-1β, TNF-α, IL-2, IL-6, IL-8, IL-17, IFN-γ, MCP-1 and RANTES in the plasma were measured using the multiplex cytokine commercial kit Bio-PlexProTM Human Cytokine Grp I Panel 27-Plex (Bio-Rad, USA).” Bonferroni correction (all p-values multiplied by 8) might be too harsh, but at least FDR correction (Benjamini-Hochberg or similar) should be applied to all comparisons. In fact, this is a minimal approach, since for each association 4 genetic models were tested in 3 situations (cases, controls and total), correcting for all these would result in no significant associations in the entire study.

However, it seems 27 different chemokines/cytokines were actually quantified with the Bioplex kit, according to the manufacturer: 

• FGF basic

• Eotaxin

• G-CSF

• GM-CSF

• IFN-γ

• IL-1β

• IL-1ra • IL-2

• IL-4

• IL-5

• IL-6

• IL-7

• IL-8

• IL-9 • IL-10

• IL-12 (p70)

• IL-13

• IL-15

• IL-17A

• IP-10

• MCP-1 (MCAF) • MIP-1α

• MIP-1β

• PDGF-BB

• RANTES

• TNF-α

• VEGF

Please mention if the cytokines/chemokines not appearing in the manuscript were undetectable (not very likely for e.g. MIP-1alpha/beta and CXCL10), or why they were not selected for analysis. I assume some of these quantifications were previously published (IFN-gamma, IL1beta, IL1RA,…), so it would be correct to state this as a reanalysis.

2. Statistical reporting of the results is incorrect and often misleading:

2.1. Odds ratios are interpreted as if they were relative risk (RR): RR = 1.74 means exposed individuals are 74% more likely to be diseased, OR = 1.74 means that the odds of disease is 74% higher in exposed individuals, which is intrinsically different. For instance, in line 412: “Here, we showed that the GG genotype of the MIR499A rs3746444 is associated with a 74% increased risk of developing CL. Similarly, the G allele confers a 27% elevated risk of developing CL suggesting that the G allele may contribute to the development of Lg-CL.” 

2.2. Large CI including 1 are often arguing against the significance claimed by the authors, e.g. “Carriers of rs2910164 CC genotype have a 33% elevated risk with a 95% confidence limits ranging from less 4% to an increase of 83%” should be reported as “no significantly increased odds of disease for the CC genotype.

2.3. Raw p-values, uncorrected for age and gender are mentioned in the abstract, while there is a bias between cases and controls and none of the associations remain. As discussed below, the significant findings in males only also make more sense if occupational exposure is considered, this should be emphasized instead of the many weak associations.

Please correct the entire results section in the abstract accordingly: “Carriers of rs2910164 CC

60 genotype have a 33% elevated risk while individuals with the genotype GG have lower risk

61 of developing CL (OR=0.75 95%CI 0.55 - 1.0). Heterozygous GC individuals also have

62 lower risk of developing CL compared with homozygous carriers of the C allele (OR= 0.71

63 95%CI 0.52 - 0.99 p=0.04). In addition, individuals homozygous for the G allele seem to

64 have higher plasma IL-8 and RANTES. For the MIR499A rs3746444, individuals with the

65 GG genotype have a 74% increased risk of developing CL when compared to AA genotype.

66 The G allele confers a 27% elevated risk of developing CL suggesting that the G allele may

67 contribute to susceptibility to the development of Lg-CL (OR=1.27 (95% CI 1.0 – 1.6] p=

68 0.035). Individuals homozygous for the G allele have higher plasma IL-8 with a tendency to

69 also have high IL-6 and IL-17. Both MIR146A rs2910164 and MIR499A rs3746444 are

70 associated with the development of Lg-CL and this association is more prevalent in male

71 individuals.”

3. The paper does not have a “Limitations of the study” section. This should be inserted and at least include 1) the problem of multiple testing of a large number of associations, 2) the significant bias in both age and gender in cases vs controls, and 3) the assumption that all controls were exposed by merely living in the same endemic area might not hold, thus precluding a correct interpretation of the genetic associations. Other genetic studies have included DTH+ individuals which have been exposed to Leishmania In fact, the significant associations observed in males but not females might hint at different exposures by gender (mostly agricultural workers.

Minor comments:

1. Several typos and spelling/grammatical errors are found throughout the manuscript.

2. A (post-hoc) power calculation would be useful for the correct interpretation of the results, e.g. “with the available number of xx cases and xx controls and MAF of >0.15, an OR of xx (or larger) could be detected with 80% power.”

**Conclusions**

-Are the conclusions supported by the data presented?

-Are the limitations of analysis clearly described?

-Do the authors discuss how these data can be helpful to advance our understanding of the topic under study?

-Is public health relevance addressed?

Reviewer #1: Conclusions are partially supported by the data, as the authors presented a lengthy and highly speculative discussion, which should be modified. Some limitations are presented in the discussion, but it would be beneficial to list them separately, rather than embedded in the discussion.

Reviewer #2: Authors studied the SNPs in miR1461 and miR499 in Leishmania guyanensis CL patients. The manuscript Bring a new View point to understand the function of the SNP in the miRNAs, a molecules that was found impacting in the host-parasite interaction, correlating with suceptibility or resistance to infection.

Reviewer #3: Conclusions, as presented in the abstract, are not supported by the data (as detailed above) and should be rephrased, with an emphasis on the findings in males. A "limitations of the study" section is lacking. Public health relevance has been thoroughly addressed for cutaneous leishmaniasis.

**Editorial and Data Presentation Modifications?**

Reviewer #1: - Please see nomenclature for chemokines (RANTES and MCP1 no longer used).

Reviewer #2: (No Response)

Reviewer #3: (No Response)

**Summary and General Comments**

Reviewer #1: The manuscript by Mesquita et al. evaluated the association between genetic variants of miR146-a and miR499a and cutaneous leishmaniasis (CL) caused by L. guyanensis. They evaluated the frequency of the variants rs2910164 and rs3746444 in 850 cases and 891 controls by RFLP. In addition, the authors evaluated the expression of selected cytokines using Bioplex. The data showed that CC genotype for the rs2910164 is slightly associated with CL, while the GG genotype for rs3746444 is associated with a 75% higher risk of developing CL. The paper also shows the expression of selected inflammatory cytokines and chemokines, attempting to correlate miRNA variants and cytokine levels. 

The paper deals with an important subject, which is to find markers of susceptibility to Leishmania infection, a devastating disease. I feel that the manuscript should focus on the association of the variants with disease, as the experiments with cytokine levels are too indirect. The results can be combined as a short report. Please see some specific comments below:

- The authors adjusted the statistical analysis for sex and age in table 1 (Padj column). This correction did not return statistically significant differences. However, data presented in table 2, where they segregated by sex (not considering age) showed statistically significant associations. 

Does this suggest that age, in association with sex, is influencing the results? How do the authors explain that? 

- Does the combined genotype analysis (for example: occurrence of CC genotype for rs2910164 together with GG genotype for rs3746444) show a higher risk of disease development?

- Cytokine expression is influenced by several parameters (such as immunological, environmental, genetic, and epigenetic). However, while these miRNAs have been associated with consequences in cytokine expression due to their action in signaling molecules, it is not clear that the variants for these miRNA are functional regarding the expression of the miRNA itself. 

Have the authors measured the levels of miR146-a and miR499a in individuals with distinct genotypes? This would help make the argument of the functional activity of the variants and would help associate with cytokine expression.

- The fact that the variants are associated with expression of some but not all cytokines that are related to the molecules inhibited by the miRNA suggests that factors other than the variants influence cytokine expression. In addition, as mentioned above, the authors do not show an association between the variant and the expression of the miRNA. Thus, the cytokine data represents too far of an extrapolation of between the variant for the miRNA and its functional activity and because of that it should be removed.

- Discussion is too long and speculative.

Reviewer #2: I consider the work address a higher relevance to understand how the host-genetic interfere in the post-transcriptional regulation of gene expression and consequences to inflammation. I recommend a minor revision.

Minor points:

Abstract: include the main conclusion in the final of abstract.

In the introduction section: 

author can cite the original works that describe the miRNA function, as Refs: Baltimore et al., 2008; Bartel, 2004; Bartel, 2009.

The work from Geraci and others can be used to explain the impact of miRNAs in host susceptibility to infection, as REF: Lemaire et al., 2013; Geraci et al., 2015; Muxel et al., 2017,2018; Acuna et al., 2020; Tiwari et al., 2017, Nunes et al., 2018,

Kumar et al, 2018; Das et al., 2020; Souza et al.,2021 and others

- 

Are the miR-146a and miR-499 upregulated in Leishmania-infected human/mouse host to support the evaluation of miRNA polymorphisms? Inlcude some information in the introduction and discussion sections.As REFs:

- Kumar V, Kumar A, Das S, et al. Leishmania donovani Activates Hypoxia Inducible Factor-1α and miR-210 for Survival in Macrophages by Downregulation of NF-κB Mediated Pro-inflammatory Immune Response. Front Microbiol. 2018;9:385. Published 2018 Mar 8. doi:10.3389/fmicb.2018.00385 ; 

- Das S, Mukherjee S, Ali N (2021) Super enhancer-mediated transcription of miR146a-5p drives M2 polarization during Leishmania donovani infection. PLoS Pathog 17(2): e1009343. https://doi.org/10.1371/journal.ppat.1009343

-Diotallevi A, De Santi M, Buffi G, Ceccarelli M, Vitale F, Galluzzi L and Magnani M (2018) Leishmania Infection Induces MicroRNA hsa-miR-346 in Human Cell Line-Derived Macrophages. Front. Microbiol. 9:1019. doi: 10.3389/fmicb.2018.01019

Reviewer #3: A large sample size and correct design are the strengths of this study, while overselling and not correcting statistical findings are the main limitations.

PLOS authors have the option to publish the peer review history of their article (what does this mean?). If published, this will include your full peer review and any attached files.

Reviewer #1: No

Reviewer #2: Yes: Sandra Marcia Muxel

Reviewer #3: No
---

## [Decision Letter · Decision Letter 1]

7 Sep 2021

Dear Dr. Ramasawmy,

We are pleased to inform you that your manuscript 'Variants of MIRNA146A rs2910164 and MIRNA499 rs3746444 are associated with the development of cutaneous leishmaniasis caused by Leishmania guyanensis and with plasma chemokine IL-8' has been provisionally accepted for publication in PLOS Neglected Tropical Diseases.

Best regards,

Claudia Ida Brodskyn

Associate Editor

Steven Singer

Deputy Editor

The authors answered all the questions raised by the referees. and the manuscript deserves to be published. Please, note that there is a minor correction raised by the reviewer 3 the needs to be observed.

Reviewer's Responses to Questions

**Key Review Criteria Required for Acceptance?**

**Methods**

-Are the objectives of the study clearly articulated with a clear testable hypothesis stated?

-Is the study design appropriate to address the stated objectives?

-Is the population clearly described and appropriate for the hypothesis being tested?

-Is the sample size sufficient to ensure adequate power to address the hypothesis being tested?

-Were correct statistical analysis used to support conclusions?

-Are there concerns about ethical or regulatory requirements being met?

Reviewer #1: YES

Reviewer #2: The objetives are clearly delivered in the study, presentign a adequate sampleou size and statistical analysis.

Reviewer #3: Minor correction needed: please specify in the Methods section which FDR correction was used (Benjamini-Hochberg, Benjamini-Yekutieli, or another?).

**Results**

-Does the analysis presented match the analysis plan?

-Are the results clearly and completely presented?

-Are the figures (Tables, Images) of sufficient quality for clarity?

Reviewer #1: YES

Reviewer #2: The manuscript present an adequate analysis of polymorphism in MIR146A rs2910164 and miR499A rs3746444 in Lg-CL patients. The data analysis of polymorphism in MIR146A rs2910164 in Lg-CL patients correlates the allele C with higher risk to develop leishmaniasis in male, previously implicated in psoriasis. Also, authors identified a lower of plasma levels of IL-8 in Lg-CL with homozygous allele C, which can be Target by miR-146.

Reviewer #3: Yes

**Conclusions**

-Are the conclusions supported by the data presented?

-Are the limitations of analysis clearly described?

-Do the authors discuss how these data can be helpful to advance our understanding of the topic under study?

-Is public health relevance addressed?

Reviewer #1: YES

Reviewer #2: The comparative analysis of distributiion of genotypes and allelesin MIR146A rs2910164 and miR499A rs3746444 in Lg-CL patients support the conclusions presented in the manuscript.

Reviewer #3: Yes

**Editorial and Data Presentation Modifications?**

Reviewer #1: N/A

Reviewer #2: (No Response)

Reviewer #3: Minor correction needed: please specify in the Methods section which FDR correction was used (Benjamini-Hochberg, Benjamini-Yekutieli, or another?).

**Summary and General Comments**

Reviewer #1: The authors have responded satisfactorily to the comments I presented.

Reviewer #2: The manuscript bring a new approach for understand the funcional role of miRNAs during Leishmania infection by focus in the polymorphism of MIR146A rs2910164 and miR499A rs3746444 in Lg-CL patients. The allele frequências and genotypes in these genes can interfere in resistance or suceptibility to leishmaniasis, including the control of proinflammatory response to parasite.

Reviewer #3: I agree with all changes to the manuscript and endorse its publication in the present form, with the minor correction detailed above.

PLOS authors have the option to publish the peer review history of their article (what does this mean?). If published, this will include your full peer review and any attached files.

Reviewer #1: No

Reviewer #2: **Yes: **Sandra Marcia Muxel

Reviewer #3: No

---

## [Editor Report · Acceptance letter]

15 Sep 2021

Dear Dr. Ramasawmy,

We are delighted to inform you that your manuscript, "Variants of MIRNA146A rs2910164 and MIRNA499 rs3746444 are associated with the development of cutaneous leishmaniasis caused by Leishmania guyanensis and with plasma chemokine IL-8," has been formally accepted for publication in PLOS Neglected Tropical Diseases.

Best regards,

Shaden Kamhawi

co-Editor-in-Chief

Paul Brindley

co-Editor-in-Chief
